# Modelling the Unidentified Abortion Burden from Four Infectious Pathogenic Microorganisms (*Leptospira interrogans*, *Brucella abortus*, *Brucella ovis*, and *Chlamydia abortus*) in Ewes Based on Artificial Neural Networks Approach: The Epidemiological Basis for a Control Policy

**DOI:** 10.3390/ani13182955

**Published:** 2023-09-18

**Authors:** Gabriel Arteaga-Troncoso, Miguel Luna-Alvarez, Laura Hernández-Andrade, Juan Manuel Jiménez-Estrada, Víctor Sánchez-Cordero, Francisco Botello, Roberto Montes de Oca-Jiménez, Marcela López-Hurtado, Fernando M. Guerra-Infante

**Affiliations:** 1Department of Cellular Biology and Development, Instituto Nacional de Perinatología, Ciudad de Mexico 11000, Mexico; drgarteagat@yahoo.com.mx; 2Military School of Health Officers, University of the Mexican Army and Air Force, SEDENA, Ciudad de Mexico 11650, Mexico; 3Laboratory of Leptospirosis, National Centre for Disciplinary Research in Animal Health, and Food Safety (CENID-SAI, INIFAP), Ciudad de Mexico 05110, Mexico; luamvet@gmail.com; 4Laboratory of Bacteriology, National Centre for Disciplinary Research in Animal Health, and Food Safety (CENID-SAI, INIFAP), Ciudad de Mexico 05110, Mexico; hernandezandrade@yahoo.com; 5Laboratory of Molecular Biology, Public Health Laboratory of State of Mexico, ISEM, Toluca 50180, Mexico; jiesjm@gmail.com; 6Department of Zoology and National Pavilion of Biodiversity, Institute of Biology, National Autonomous University of Mexico, Ciudad de Mexico 04510, Mexico; victor@ib.unam.mx (V.S.-C.); francisco.botello@ib.unam.mx (F.B.); 7Faculty of Veterinary Medicine, Universidad Autónoma del Estado de Mexico, UAEM, Toluca 50295, Mexico; romojimenez@yahoo.com; 8Department of Infectology and Immunology, Instituto Nacional de Perinatología, Ciudad de Mexico 11000, Mexico; diaclaro2000@yahoo.com.mx; 9Department of Veterinary Microbiology, Escuela Superior de Ciencias Biológicas, IPN, Ciudad de Mexico 11340, Mexico

**Keywords:** machine learning, *Leptospira* spp., smooth *Brucella* spp., *Brucella ovis*, *Chlamydia abortus*, zoonoses

## Abstract

**Simple Summary:**

Since the beginning of the Cenozoic era, microorganisms have circulated worldwide, many of them cause significant morbidity and mortality in animals and humans. Ecological changes may favor transmission, and modifying of host–environment/pathogen interactions and leptospirosis is a good example of this, as it evolved from pathogens circulating in wildlife. Using data generated from an epidemiological survey and from the lab, the abortion burden of multiple microorganisms in sheep was predicted according to the artificial neural network approach and Generalized Linear Model (GLM) in a geographic area of the Mexican highlands. The results showed that the best GLM is integrated by the serological detection of *Leptospira interrogans* serovar Hardjo and *Brucella ovis* in animals on the slopes with elevation between 2600 and 2800 masl in the municipality of Xalatlaco. The sheep pen built with materials of metal grids and untreated wood, dirt and concrete floors, bed of straw, and the well water supply were also remained independently associated with infectious abortion. We suggest that sensitizing stakeholders on good agricultural practices could improve public health surveillance.

**Abstract:**

Unidentified abortion, of which leptospirosis, brucellosis, and ovine enzootic abortion are important factors, is the main cause of disease spread between animals and humans in all agricultural systems in most developing countries. Although there are well-defined risk factors for these diseases, these characteristics do not represent the prevalence of the disease in different regions. This study predicts the unidentified abortion burden from multi-microorganisms in ewes based on an artificial neural networks approach and the GLM. Methods: A two-stage cluster survey design was conducted to estimate the seroprevalence of abortifacient microorganisms and to identify putative factors of infectious abortion. Results: The overall seroprevalence of *Brucella* was 70.7%, while *Leptospira* spp. was 55.2%, *C. abortus* was 21.9%, and *B. ovis* was 7.4%. Serological detection with four abortion-causing microorganisms was determined only in 0.87% of sheep sampled. The best GLM is integrated via serological detection of serovar Hardjo and *Brucella ovis* in animals of the slopes with elevation between 2600 and 2800 meters above sea level from the municipality of Xalatlaco. Other covariates included in the GLM, such as the sheep pen built with materials of metal grids and untreated wood, dirt and concrete floors, bed of straw, and the well water supply were also remained independently associated with infectious abortion. Approximately 80% of those respondents did not wear gloves or masks to prevent the transmission of the abortifacient zoonotic microorganisms. Conclusions: Sensitizing stakeholders on good agricultural practices could improve public health surveillance. Further studies on the effect of animal–human transmission in such a setting is worthwhile to further support the One Health initiative.

## 1. Introduction

Decisions to initiate a public health program in the community are largely based on accurate estimates of the burden of disease. Abortion in sheep, including early fetal loss and stillbirth of lambs, is a major cause of economic loss for farm workers and farming communities. Naturally, abortion can be a common source of infection in humans and therefore of public health importance when caused by zoonotic microorganisms [1]. While many infectious causes of abortions occur worldwide, etiologic data based on robust laboratory-confirmed diagnoses are subject to studies of farming systems in high- and middle-income countries with demographic characteristics not observed in low-income settings [2]. A comparison of etiologies between countries and geographic areas is also not feasible for use in public health policy since epidemiological studies vary in methodologies and in the prevalence of the specific pathogens that are investigated. Therefore, it is necessary to implement methods that balance the economic costs allocated to the programs by increasing the accuracy and feasibility (efficiency) of the sampling [3].

In Mexico, as in other developing countries, data on the prevalence of zoonotic infectious agents are generally obtained through cross-sectional studies. Due to the limited availability of sampling frames and the high costs of transport to the sampling site, it is impossible and impractical to select a simple random sample (SRS) of the animals in the population solution for most of these studies is to obtain a cluster sample by randomly selecting flocks from a checklist and then randomly reselecting a defined number of animals within each herd [4]. In addition, new epidemiological surveillance tools, such as artificial intelligence and satellite geoprocessing, will greatly complement public health systems [5,6].

The relationship between infectious agents and host to human transmission is influenced by many environmental factors. *Leptospira*, *Brucella*, and *Chlamydia* are some of those endemic infectious agents in many countries without the availability of epidemiological surveillance systems or adequate diagnostic laboratories [7,8]. Rural areas tend to be a higher risk compared to urban areas due to a larger number of animal reservoirs in agricultural and forestall areas, as well as a higher level of transmission between domestic and wild animals [9,10]. On the contrary, urban leptospirosis, for example, is relatively easier to control through the implementation of anti-epizootic measures such as controlling the reproduction of rats by avoiding their availability of food and shelter [11]. A dirty environment and rodents will always be associated with the transmission mechanisms of leptospirosis due to the possible presence of garbage and contaminated water and soil [12,13,14]. Many studies have reported that rural areas with limited access to clean drinking water and sanitation are more conducive to human infection [15]. Furthermore, leptospirosis has been identified as an occupational disease where humans acquire the infection mainly through exposure to livestock, agricultural, and military activities [16].

Artificial intelligence algorithms can be used for interdisciplinary applications, such as configuring robotic systems or validating biological systems that provide solutions to complex real-world problems in the current era of digital healthcare systems [17]. Predictive learning tries to build suitable prediction rules using the learning algorithms specific only by processing the data without any knowledge of it. Relevant information during problem solving is supposed to be contained in the available data, and it is the responsibility of the learning algorithm to automatically extract and organize this information to obtain a prediction rule.

In artificial neural network models, the artificial neuron simulates the essence of neural biological systems by imitating their behavior. Each neuron takes in input from other neurons and processes it using an activation function to pass on the output to the next neurons, and each information passes around through a connection that has a specific strength or weight equivalent to a biological neuron’s synaptic efficiency. Each artificial neuron also has a particular threshold value, and the difference between the weighted sum of the inputs and the threshold value comprising the neuron’s activation (post-synaptic potential). The neuronal activation signal passing through an activation or transfer function (non-linear algebra component) produces the neuron’s output. This non-linear activation function limits the range of values that the output variable of a neuron can take. Therefore, each neuron’s activity level is a function of inputs it receives and a result that is sent as a signal through its connections with other neurons [18].

Understanding the risk of infection in agropastoral settings where herds mix with each other and where different pathogens coexist with ease of transmission to humans living with sheep remains a major challenge. Field studies in this area have reported risk factors for within-flock transmission of *Leptospira* in the valley region, as well as detection of *Chlamydia abortus* via molecular testing in fetuses and abortive products in sheep [19,20]. The purpose of the study is to estimate the unidentified abortion burden from *Leptospira* serovars, smooth *Brucella* species (smooth *Brucella* spp.), *Brucella ovis* (*B. ovis*) and *Chlamydia abortus* (*C. abortus*), as well as the identification of putative factors of abortion in sheep. This is intended to determine risk areas to identify possible new outbreaks towards the development of a regional zoonotic disease surveillance program. 

## 2. Materials and Methods

### 2.1. Ethical Considerations

Approval for conducting this study was obtained from the Institutional Committee for Research and Advanced Studies at the UAEM Animal Health Center, Toluca, Mexico, whose protocol number was 2230/2006U. We analyze data collected from a field survey as well as serological screening of leptospirosis, brucellosis, and ovine enzootic abortion of unvaccinated ewes, and flocks were the primary sampling unit. Sampling was performed from January to December 2018. Location of sites of grazing sheep under traditional silvopastoral system with communal use of land resource required global positioning system equipment and a digital camera to take pictures of the surrounding grazing areas. To reduce the risk of injury and death in sheep, as well as to guarantee the safety of keepers and researchers, the good practice guides, standards, and recommendations issued by Teagasc—Agriculture and Food Development Authority were followed [21].

### 2.2. Owners’ Participation

To facilitate unbiased estimates, a two-stage cluster survey design was conducted. In the first stage, all sheep owners belonging to the Regional Association of Sheep Breeders were invited to participate. All owners were registered with a consecutive registration number. Collaboration for this research was encouraged through participation in educational forums with a focus on health promotion and prevention of zoonotic diseases. The confidentiality of the results of the study and offer free serologic testing to producers that were selected were agreed. The owners of flocks were randomly selected using the lottery method. An invitation letter and a proposed sampling schedule were sent to each of the 35 selected owners. The farmers answered a set of questions regarding herd structure, land use, animal production, flock size, breeding, drinking water, health trait, animal production, performance of activities in lambing sheep, farm building and equipment, and farmer profile topics (see Appendix A for questionnaire). The questions were answered during a personal interview with the owner or service farm workers, always avoiding mentioning economic aspects. Samples for serological screening were collected, and the exact geographic distribution of the animal groups was demarcated. We obtained verbal consent from the owners/farmers due to the high level of trust.

### 2.3. Geographic Distribution of Selected Groups

To identify the geographic location of the animal groups and the areas where the sheep graze, a satellite positioning system (Global Positioning System, GPS; Magellan Meridian, Thales Navigation, San Dimas, CA, USA) was used. The coordinates of latitude, longitude, and altitude of the position, time, and satellites of each of the animal groups selected herds were recorded in the field log. The signal received from three satellites was considered valid. A geographic information system (GIS) was developed to store and analyze of geographical and spatial data for the study area. Climatic and environmental data, including topography, hydrography, average annual precipitation, and maximum and minimum temperatures recorded, as well as the distribution of human communities, were obtained from INEGI-2017 [22], and they are summarized in Appendix A. A map of study the site was build using QGIS, version 3.0 (QGIS.org; https://qgis.org/es/site/, accessed on 18 March 2023). The vectorial layer of the municipalities was based on INEGI-2005 [23], and Google Satellite and Terrain platforms, respectively. The elevational gradient (2572–2995 masl) was established along the southeastern slopes of the State of Mexico in central Mexico, where animal sampling was conducted at thirty-five study sites within elevational belts of approximately 200 m each.

### 2.4. Serological Screening

In the second stage of sampling, it was necessary to randomly select an initial animal and then another 9 more animals in each of the first-stage clusters. Blood samples from 345 ewes of reproductive age were collected. The blood samples were obtained through puncturing of the jugular vein. The serum samples were obtained by centrifuging the test tubes containing the blood samples at 1000× *g*. The samples were then kept frozen at −20 °C until use.

To detect the presence of abortion-causing microorganisms in ewes and obtain a preliminary picture of their epidemiology, a serological survey for antibodies against *Leptospira* spp., smooth *Brucella* spp., *B. ovis*, and *C. abortus* was carried out. Lytic/agglutinating activity of antibodies against Leptospira serovars was performed using a microscopic agglutination test (MAT) as described in the OIE Diagnostic Tests and Vaccines for Terrestrial Animals [24]. Antigens used in the MAT included ten serovars usually detected in this area. For the study panel, the H-89 (Hardjo genotype hardjoprajitno), Sinaloa ACR (Portland-vere), and Palo Alto (Icterohaemorrhagiae) strains isolated in Mexico and kindly classified for CA Bolin from Ames, IA, USA, were included. The cut-off titer (≥1:100) was considered positive. The end-point titer was the highest serum dilution showing agglutination of at least 50% of the leptospires—it was also included in MAT, positive reference controls for each strain.

The detection of anti-smooth *Brucella* spp. antibodies was analyzed with a rapid serum agglutination test (Rose Bengal Plate Test, RBPT) using *B. abortus* biovar 1 strain 1119-3 antigen with cellular concentration at 3% that covers the reactivity of *B. melitensis* and *B. abortus*. The RBPT result showed only two classifications: (a) positive reaction and (b) negative reaction depending on the presence or absence of agglutination. The sensitivity and specificity of RBT calculated for diagnosing *Brucella* in goats are 59 % and 98%, respectively [25]. 

The Oüchterlony double immunodiffusion (AGID) test was used to detect anti-bodies against *B. ovis* [26]. The appearance of precipitation lines of complete identity with the control serum lines was considered positive serum for antibodies against *B. ovis* detection. The absence of precipitation lines of complete identity was considered as a negative result. The AGID antigens are highly sensitive (70.1%) and specific (100%) for the serological diagnosis of *B. ovis* infection in rams [27]. 

Also, serum samples were analyzed for detection of antibodies against *C. abortus* using a recombinant commercial ELISA test (*Chlamydia abortus* serum verification, P00700/05-25/05/04. Institute Pouquier, Montpellier, France) used in accordance with the manufacturer’s instructions. The ELISA assay for diagnosing *C. abortus* infection in sheep has shown sensitivity and specificity of 93.5% and 98.5%, respectively [28]. 

### 2.5. Statistical Analysis

The overall seroprevalence of each one of the four abortion-causing microorganisms and the 95% confidence interval (CI) were calculated based on the following equation:(1)$$\hat{p}=\frac{N}{Mn}\sum_{i=1}^{n}y_{i}$$
where y_i_ = n_i_ (d) is the number of animals sampled and proportion of disease-positive animal proportion in the cluster, N is the number of clusters, n_c_ is the number of cluster sampling, M_i_ is the number of animals in cluster, and ρ_h_ is the prevalence amongst flocks of the population.

The variance estimated for number of clusters is given by:(2)$$\hat{v}\left(\hat{p}\right)=\frac{N^{2}}{M^{2}}\left(1-\frac{n}{N}\right)\frac{{\hat{s}}_{b}^{2}}{n}$$
(3)$${\hat{s}}_{b}^{2}=\frac{1}{N-1}\sum_{i=1}^{N}{\left(y_{i}-\bar{y}\right)}^{2}$$

All categorical demographics and characteristics from sheep groups were expressed as weighted percentages with 95% confidence intervals (CI). In all cases, the analysis (two-sided test) was considered statistically significant at *p* < 0.05. For continuous variables, a statistically significant difference was determined using Student’s *t*-test or the Mann–Whitney U test in the case of nonnormality. Categorical data were evaluated using Fisher’s exact test, and correlations between categorical and continuous variables were examined using Spearman’s rho. The Kolmogorov–Smirnov test was used to verify data normality.

#### 2.5.1. Artificial Neural Networks Model (ANNM)

The statistical patterns recognition from machine learning was performed with multilayer perceptron algorithm. The creation of topology and training of the network from various combinations of variables obtained from data of field survey and serological status of microorganisms considered the hidden layers, training cycles, and the parameters of the mathematical training function. The statistical properties of the training (70%), validation (20%), and test (10%) data of the ANNM allowed the prediction and classification of variables. Prior to exporting the data matrix to the ANNM for training, normalization of the data was performed to restrict the data range within a 0 to 1 scalar since the sigmoid activation function was assigned for each neuron in the middle layer. The weight of 0.5 corresponding to the activated value allows a better neural response. Also, categorical variables were recoded into “dummy” variables to improve response.

The performance criteria were decided based on the system-estimated values that occurred during training. The prediction accuracy was estimated from the area under the curve obtained by the receiver operating characteristic curve (ROC). A plot of power as a function of the type I error of the decision rule illustrated the performance of the binary model as its discrimination threshold is varied.

#### 2.5.2. Generalized Linear Model (GLM)

The relationship between abortion and risk factors, protective or confounding, was determined by preprocessing variables obtained using the multilayer perceptron neural network model. We used a type of GLM and assumed that ewe abortion followed a binomial distribution. The use of logit link function facilitated the estimated contribution from covariates to be additive. A covariate selection algorithm was used to facilitate incorporation into the final GLM. To select the best model, we use a modified backward elimination procedure [29]. The model was built in three steps. (1) All covariates predicted by ANNM according to normalized significance were entered. (2) It started with a complex model verifying if the interaction terms were necessary with the elimination of the terms successively. The likelihood ratio statistic with *p* value < 0.3 suggested integration of the individual covariate. The next stage considered dropping a term from the main effects model. One by one of the covariates were eliminated from the initial model according to the highest p-value comparing the current model with the previous one. The Bayesian Information Criterion (BIC) and Akaike (AIC) index were used to choose between two or more alternative GLMs. (3) A receiver operating characteristic (ROC) curve was used as plot of sensitivity as a function of (1—specificity) for evaluating predictive power of all possible models. Statistical analyses were performed with Epi-Info v7.2.4.0 software (Centers for Disease Control and Prevention, CDC: Atlanta, GA, USA), and IBM SPSS v25 software (IBM Corp: Armonk, NY, USA).

## 3. Results

### 3.1. Places of Sampling

The study was carried out in a lacustrine zone of the trans-Mexican neovolcanic belt to the south-east of the State of Mexico, Mexico (19° N; 99° W). Based on the GPS readings recorded, the sampling sites were in three altitudinal zones of the mountainous region: zone 1 (elevations up to 2600 masl); zone 2 (2601 to 2800 masl); and zone 3 (>2800 masl) (Appendix A; Appendix A). The map of the sampling sites and the geographical coordinates of study area is shown in Figure 1.

### 3.2. Antibody Prevalence of Multi-Pathogens and History of Abortion

The 345 sera available from sheep groups were screened for causing abortions multi-microorganisms. An infectious etiology was determined in 83.8% (289/345) of the ewes with a history of abortions in which the serological diagnoses were determined. Overall seroprevalence of smooth *Brucella* spp. was 70.7% (95% CI 65.7–75.8), while *Leptospira* spp. was 55.2% (95% CI 46.9–63.4), *C. abortus* was 21.9% (95% CI 5.3–38.5), and *B. ovis* was 7.4% (95% CI 0.0–28.1) according to the RBPT, MAT, ELISA, and AGID tests, respectively. The weighted seroprevalence of multiple organisms was only higher for smooth *Brucella* spp., but not for *Leptospira* spp., *B. ovis*, and *C. abortus* in aborted ewes. The weighted seroprevalence of smooth *Brucella* spp. was higher in aborted animals than in animals without a history of abortion (83.3% vs. 67.7%, respectively) (*p* < 0.007). Concerning seroprevalence by geographic regions, only seroprevalence of *C. abortus* for those aborted ewes from slopes with elevations up to 2600 masl was higher than for aborted ewes from slopes with elevations above 2800 masl (75.8% vs. 41.7%, respectively) (OR = 4.4; 95% CI 1.2–15.9; *p* < 0.02).

The probability of abortion due to *Brucella* infection with a positive-RBPT test was estimated to be around 98% (uncertainty coefficient 0.02), while *C. abortus* infection with a positive ELISA test was 82% (uncertainty coefficient 0.18). All sheep groups showed seropositivity to smooth *Brucella* spp., while 94.3% (33/35) of the investigated flocks showed antibodies against *Leptospira* spp., *C. abortus*, and *B. ovis*. At the individual level, the seroprevalence with two and three abortifacient microorganisms in the sampled sheep was higher from slopes with elevations up to 2600 masl than animals from slopes with elevations above 2600 masl. Serological detection with the four abortion-causing microorganisms was determined only in 0.87% (3/345) of sheep sampled. Table 1 shows the seropositivity of *Leptospira* spp., smooth *Brucella* spp., *B. ovis*, *C. abortus* antibodies in the ewe population sampled varied by altitudinal zonation. In contrast, Table 2 shows the weighted seropositive of *Leptospira* serovars in ewes sampled with history of abortion. Antibodies against *Leptospira* were detected in the sera of 188 ewes. The MAT results only showed questionable titers in 5.3% (166/3,105) of the total reactions (1:50 reactivity in sera that tested negative). The prevalence of agglutinins was detected in the sera of aborted ewes, mostly against serovar Icterohaemorrhagiae (37.2%, 95% CI 31.5–42.9), Bratislava (27%, 95% CI 21.8–32.2), and Canicola (12.3%, 95% CI 8.5–16.1). Nevertheless, serological profiles of the *Leptospira* serovars of the abortive and nonabortive animals, taken individually throughout the study, were only significantly different for serovar Grippotyphosa (*p* < 0.003). All sera that tested positive had antibody titers from 1:100 to 1:6400, with the strongest reactions to type strain Hond Utrecht IV and Portland-vere (Figure 2).

### 3.3. Validation of the Multilayer Perceptron Algorithm for Abortion in Ewes

The structure of infectious abortion prediction model based on the machine learning of multilayer perceptron algorithm is shown in Figure 3. The multilayer perceptron model allowed us to identify the patterns of predictive variables among the universe of non-linear variables related to abortion in sheep after training and the validation of the neural network. The neural network architecture showed a minimum number of hidden neurons to avoid the problem of overfitting, with which the perceptron algorithm obtains the best performance (Figure 3a), while the evaluation of the performance of the multilayer perceptron algorithm based on the area under the curve of the ROC curve showed that 80% of the ewes with a history of abortion were correctly classified (Figure 3b). The relative importance of input neurons for the prediction of abortion in sheep obtained from the multilayer perceptron model (Figure 4). The importance plot showed that the results are dominated by mixed infections with *Leptospira* spp.-*C. abortus*-smooth *Brucella* spp.-*Brucella ovis* (100%), followed by whether *Leptospira* serovar Canicola was identified exclusively (79.3%), with other infection predictors following far behind. The importance of mixed infections was clearly visible in the chart and much less visible for smooth *Brucella* spp. infection.

### 3.4. Determinants of Infectious Abortion in Ewes

The relationship between infectious abortion and preprocessing variables obtained by the multilayer perceptron neural network model showed 28 possible GLM models that can support prevention and control activities in different geographical areas (Appendix A). The best model for infectious abortion, according to BIC and AIC, involving all four infectious pathogens showed several leptospiral serovars with a significant effect (Table 3). Serological detection of leptospirosis, which includes serovar Hardjo, as well as brucellosis caused by *B. ovis*, have a positive impact on sheep abortion (*p* < 0.001). On the other hand, the identification of serovar Grippotyphosa and Tarassovi had a significant negative effect on the outcome. The GLM additionally revealed that from the 30 analyzed exposure factors, 14 (46.7%) contribute significantly to ovine abortions. The determinants obtained from the generalized linear model were significantly different from zero, and the deviance residuals showed good model fit due to the ROC curve result (AUC: 0.89, 95% CI: 0.85–0.93; *p* < 2.94 × 10^−20^). The AUC showed the probability that a randomly chosen aborted sheep would classify higher than a randomly chosen non-aborted sheep. 

## 4. Discussion

Among the etiologies of abortion in sheep there are the presence of infectious agents, including many zoonotic microorganisms, and non-infectious causes such as nutritional, genetic, hormonal, toxic, and clinical (trauma, dystocia, prolapse) [30]. The results of our study demonstrate the infectious etiology in 82.6% (285/345) of the ewes with a history of abortions in which the serological diagnoses were determined.

It is widely accepted that seroprevalence of infectious agents causing abortion in small ruminants may vary according to geographic region, and zoonotic pathogens such as *Leptospira spp.*, smooth *Brucella spp.*, *B. ovis*, and *C. abortus* are among the most important microorganisms [31,32]. Although notification to the World Organization for Animal Health (OIE) is mandatory for many abortive pathogens of sheep due to restrictions on international trade, in Mexico, there is a lack of an effective epidemiological surveillance system that allows the development of strategies to prevent and control reproductive losses due to abortions and stillbirths of lambs, as well as to assess the prevalence of endemic diseases according to the diversity of ecosystems, with the subsequent reduction in risks to public health.

### 4.1. Towards the Strengthening of Regional Surveillance

The overall seroprevalence of microorganisms causing abortion in sheep appears to be very high, with 70.7% of animals testing positive for smooth *Brucella spp*. (*B. melitensis*, *B. abortus*, and *B. suis*) and 55.2% of animals positive for *Leptospira spp.*, followed by the seroprevalence of *C. abortus* (21.9%) and *B. ovis* (7.4%). These findings show previous exposure is strikingly higher compared to previously reported seroprevalences of brucellosis in countries such as Iran (29.1%) and Egypt (16.3%), and less than 1% in the Arabian Gulf region, countries characterized by desert climates in summer and mild in winter [33,34,35]. *Leptospira* serovar-specific antibodies have been detected in 24.7% of ewes with a history of abortions in Brazil, 8.5% in Iran, and 4.5% in Italy [36,37,38].

Leptospirosis and brucellosis are the most widespread neglected diseases throughout the world, except Antarctica [39]. Climate changes, changes in ecological niches, and the appearance of new potential maintenance hosts could represent the most important factors involved in the epidemiology of abortifacient microorganisms. The environmental and geographical characteristics of Southern region of the State of Mexico can be considered as the optimal conditions for *Leptospira* spp. and *Brucella* spp. spreading among sheep and other animals, and eventually among humans.

The high seroprevalence of leptospirosis and brucellosis in ewes is not consistent with the small number of cases of human leptospirosis and brucellosis in the State of Mexico, Mexico. Mexican population data from 2012–2022 showed 17 human confirmed cases with *Leptospira* positive serological reaction and 655 confirmed cases with *Brucella* were recorded by the Mexican Ministry of Health [40]. Owing to the lack of diagnostic laboratories and a limited reporting system, leptospirosis and brucellosis are one of several neglected diseases in Mexico, and this may be one of the reasons why few cases were identified over this period, despite the high carriage of multi microorganisms in animals. The occurrence of human leptospirosis cases is more common in the tropics, especially in South America and Asia [41,42], and in regions where brucellosis is endemic, deleterious effects are seen in both humans and domestic animals in the developing nations of Africa, South/Southeast Asia, and Latin America [43]. The appearance of zoonotic disease in new localizations, as well as the sources of transmission between wild and domestic animals, is of great importance in terms of the epidemiological dimension. For many years, small ruminants had been considered as accidental hosts of leptospires, but several studies have shown that leptospiral infection in goats and sheep is common, and these species can also act as only maintenance hosts for serovars and carriers of leptospires eliminating the microorganisms on the environment for long time periods [44]. The maintenance hosts tend to be infected by serovars that colonize the kidneys and are shed in the urine. This hosts may act as chronic selective carriers of Leptospira serovars in a range of ecosystems and possibly transmit the pathogen to accidental hosts [45]. Detection of serovar Canicola 20.3% of animals sampled and Portland vere-type strain in 3.5% observed in our study suggests the presence of a selective host such as dogs that can cause infection in sheep and possibly cause accidental infections in humans. It should be noted that the clinical differences of the disease in dogs are based on the signs associated with non-icterogenic Canicola serovar like that observed in humans as "Stuttgart disease" [46,47,48]. Leptospirosis and brucellosis in sheep pose major threats to public health from direct contact with infected animals or their contaminated biological secretions (e.g., Amniotic fluid or vaginal discharge, and aborted fetuses or placentae), as well as the consumption of meat, unpasteurized milk, and dairy products produced with consequent economic loss from restrictions on contaminated dairy products [49].

Previous epidemiological investigations reported the circulation of *Leptospira* serovars in this mountainous region with an overall seroprevalence of 54.5% and detection the most likely infecting serovars as Icterohaemorrhagiae (54.5%), Bratislava (40%), Canicola (19%), and Tarassovi (15.8%) [19]. In this study, the overall seroprevalence of 55.2% is consistent with previously reported; but serological detection against the serovars as Pomona (5.2%), Grippotyphosa (3.8%), Pyrogenes (3.5%), and Portland vere-type strain (3.5%) suggest the possibility of investigating new serovars from wild reservoirs or sheep of other environmental settings. Wild rodents are the main reservoirs for pathogenic *Leptospira* species as serovar Grippotyphosa, which cause leptospirosis in sheep [50]. Transmission of *Leptospira* serovars requires a continuous enzootic circulation of the organism between animals, although with the possibility of introduction of new serovars from animal reservoirs, both wildlife and domestic animals [44]. According to Guedes et al. [51], the microscopic agglutination test is a good serological technique for the detection of antibodies against *Leptospira* serovars, but cross-reactions and paradoxical reactions are frequently observed with MAT. Serological paradoxical reactions and cross-reactions between serogroups were observed using MAT in our study, but the presence of high-titer *Leptospira* seropositivity (>1:200) in 36.6% of Bratislava seropositive sheep, 31.3% of Icterohaemorrhagiae, 14.5% of Canicola, 4.9% of Tarassovi, 2.0% of Grippotyphosa, 1.4% of Pomona, and 1.3% of Pyrogenes suggests the possibility of infection with these serovars. Antibody titers >1:100 detected in these animal sera probably resulted in an overestimation of overall seroprevalence of leptospirosis.

### 4.2. Epidemiological Control of Unidentified Abortion

Data obtained in our study have allowed the successful implementation of an ANNM to model mixed infection of four-abortive agents in sheep with a history of abortion as in other epidemiological studies [52,53]. Compared to stochastic models, the multilayer perceptron algorithm provided adequate prediction of abortion cases without the need for prior statistical correlations, or the assumptions required by common epidemiological models. A multilayer perceptron is a neural network that learns the relationship between linear and non-linear data and is considered an easy tool for the prediction of different diseases. The multilayer perceptron algorithm facilitated the identification of the statistical patterns among the infinite non-linear combinations related to abortion in sheep after training and validation. This is the first article in a series of possible ones that will use deep machine learning in the prognosis of diseases in animals. Deep learning has gained attention in recent decades for its innovative application in areas such as image classification using only pixels and labels as input layers, speech recognition, and automatic translation of text from one language to another without human involvement.

Methodological strategy based on machine learning algorithms allowed the identification of the preprocessed variables associated with abortion in sheep. The percentage of predictive values in the training and test performance in the aborted sheep classification from the multilayer perceptron algorithm was 89.4% and 88.2%, respectively. The adequate performance of the algorithm was obtained by the ROC curve that demonstrated an area under the curve to correctly predict abortion in sheep of 86.2%. Based on machine learning, the normalized importance values were obtained, which served to integrate the variables of GLM initial model. The final GLM appeared to fit the data well (overdispersion coefficient statistic = 0.83). If the value of the overdispersion coefficient is >1, this would show that meaning the variance is much larger than the mean, and so the GLM model is not appropriate. The area under the ROC curve (0.89) was significantly different from 0.5, since the *p* value was <0.001, indicating that the GLM classified the group of aborted ewes significantly better than chance. The final GLM showed a high predictive capacity (89%); in other words, 307 of the 345 sheep sampled were correctly classified. The result obtained via GLM allows us to know the exact extent of the abortion burden of zoonotic diseases in the region of the trans-Mexican neo-volcanic belt. The detection of serovar Hardjo and *Brucella ovis* in animals of the slopes with elevation between 2600 and 2800 meters above sea level from the municipality of Xalatlaco were independently and significantly related to an increased risk of abortion. The water well supply, sheep pen built with materials of metal grids and untreated wood, dirt and concrete floor, and bed of straw were also independently associated to a risk of abortion. The results of our study provide important data for use in regional public health policy. The strengthening of epidemiological surveillance and risk assessment in these remote rural areas will allow the optimal implementation of prevention measures in the first line of health services where people are cared for. Physicians and those with less experience in remote rural areas also need to recognize the diseases of animals that affect humans through regular and structured training and supervision on identifying variations that could indicate novel outbreaks of zoonotic diseases in the human population. There is a paradox that often less-experienced physicians, nurses, and midwives are deployed to rural settings, which are viewed as undesirable, and these new graduates are provided with little ongoing support or mentoring.

### 4.3. Strength and Limitations of Study

The strength of our study includes complete information on the management of the herd and the individual animal based on the factors or characteristics that have been related to abortion in sheep, as well as the laboratory results of the serological samples obtained. Epidemiological indicators of seroprevalence of microorganisms that cause abortion in sheep to achieve its goal of providing factual, objective, reliable, and comparable information with a high precision based on clustered sampling. Also, the main factors and the less important factors in the prediction of abortion in sheep are reported by artificial intelligence learning algorithms using the multilayer perceptron model. The limitations of our study were those related to the detection of other foodborne zoonoses such as *Salmonella abortusovis*, *Campylobacter * spp., *Toxoplasma gondii*, *Listeria* spp., *Coxiella burnetii*, and *Yersinia pseudotuberculosis* which can be disseminated among animals causing abortion, as well as contaminate vegetables and fruits for human consumption. Although studies have shown that *T. gondii* has been recognized as a major cause of infectious sheep abortion in New Zealand, Australia, the United Kingdom, Norway, and the United States, the presence of the disease is usually sporadic, and serological testing is not specific for the detection of viable Toxoplasma oocysts. Evidence for abortion has been based on the detection of the DNA of *T. gondii* from fetal tissues, but these findings do not show a relationship with maternal serological results [54]. Other studies show congenital transmission from the molecular detection of *T. gondii* in the brain from aborted fetuses with low maternal seroconversion; however, in fetal brain samples from negative lambs, high maternal serology is observed [55]. The usefulness of detecting antibodies for the diagnosis of *T. gondii* in sheep abortion is currently ambivalent and controversial because there may be cross-reactivity with oocytes from other parasites, so more studies are required in this regard. Unfortunately, the problem of diagnosis of *C. burnetti* was not addressed due to the limited commercially available serological tests for using in animals. A purposeful search for *C. burnetii* infection with appropriate serological testing is required to understand the course of the disease in small ruminants, its epidemiology, and the risk of transmission to humans. Since there are few reports in Mexico in both animal and human populations that demonstrate reliable evidence of the presence of *C. burnetii*, this may not reflect the current situation of emerging rickettsial diseases [56]. Q fever, so far, is a disease considered exotic in our country.

Other difficulties were related to the budget; however, the application of the cluster sampling design balanced the feasibility of the abortion research project with the precision of the epidemiological impact measures, complemented by the data analysis based on the machine learning algorithm.

## 5. Conclusions

Our neural network approach has been able to identify multiple microorganisms and putative predictors of infectious abortion in sheep. This report includes the measurable factors to be monitored in future epidemiological studies to improve public health surveillance. Since most abortion can be prevented by the sanitation measures identified here, it only remains to propose the use of existing or currently under development vaccines to reduce the risk of animal–human transmission of zoonotic infectious agents [57]. The results of the study are expected to help establish priorities and to tailor specific public health interventions, including vaccination, to the etiology of infectious abortion in sheep from the mountainous region of Mexico.

## Figures and Tables

**Figure 1 animals-13-02955-f001:**
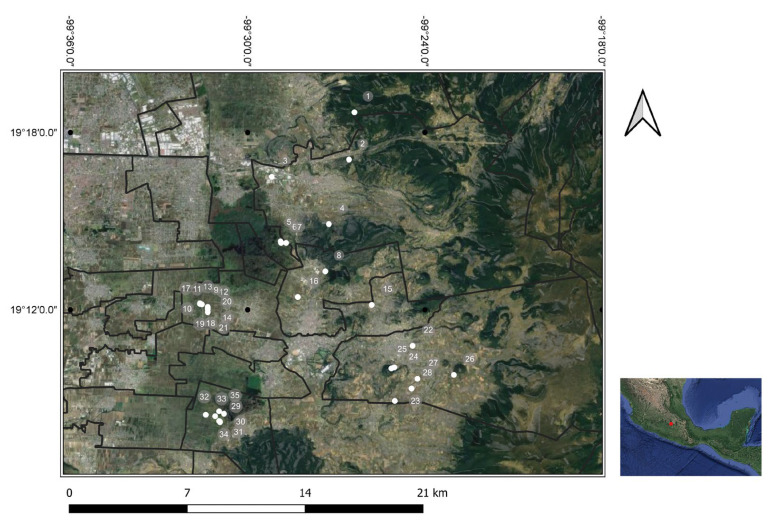
Location of selected sites sampling in the ewe abortion study (numbers). The circle points represent the georeferenced locations of the animal groups at the sampling site. A layer of geographical point features (white circle dots) represents the sampling sites where Leptospirosis, Brucellosis, and Ovine Enzootic Abortion are prevalent. The altitudinal zonation of the study site in the southeast of the State of Mexico in central Mexico, and the municipality boundaries are included in the map.

**Figure 2 animals-13-02955-f002:**
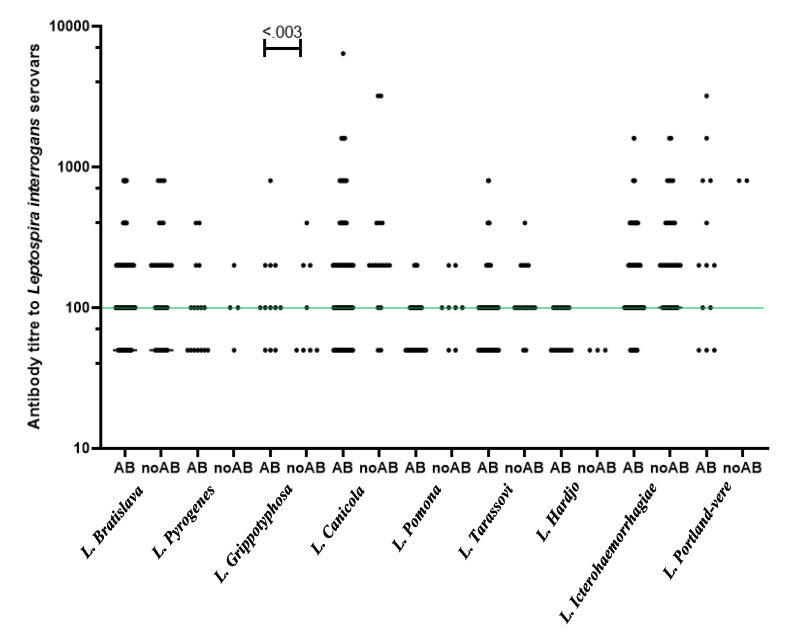
Comparison of serological MAT titers for aborting (AB) and non-aborting (noAB) ewes for antibodies against *Leptospira* serovars. Continuous line (in green) indicates the cut-off titer (≥1:100) was considered positive. *p*-value calculated by Mann–Whitney test.

**Figure 3 animals-13-02955-f003:**
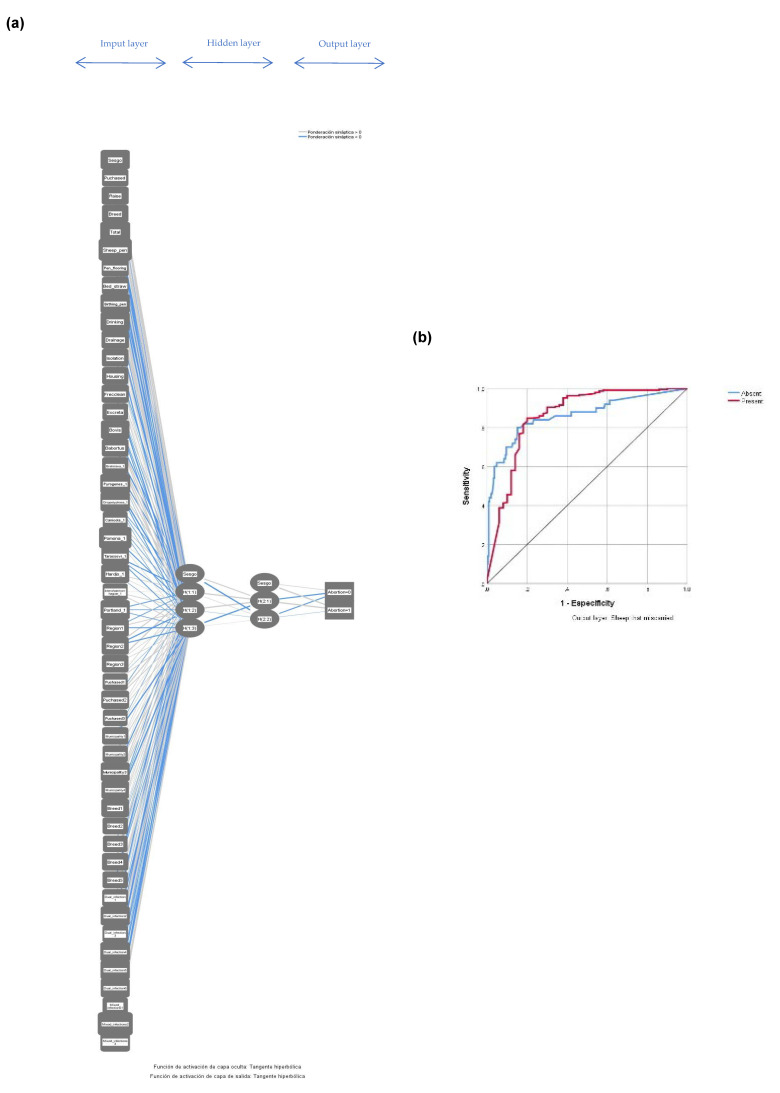
The ANNM for ovine abortion prediction. The final architecture of the model was 50 neurons in the input layer, 7 neurons in the two-layer, and 2 neurons in the output layer (Abortion: 0: absent; 1: present) (**a**); Area under the curve (AUC) for validation of the multilayer perceptron model in sheep abortion based on ROC curve analysis (**b**). Sesgo: Bias; Raise: What do you raise sheep for? 1: market, 2: breeding program; Sheep-pen: What are the materials you use to build your sheep pen? 0: bricks; 1: metal sheets and untreated wood; Sheep pen flooring: 1: dirt pen flooring only; 2: dirt and concrete pen flooring; Bed_straw: Did the ewe give birth where they bed of straw down? 1: yes; 0: no; Birthing_pen: Where was the lamb of this ewe born? 1: birthing pen; or 0: meadow; Drinking: The water supply for the animals: 1: drinking water; 2: irrigation canal; Drainage: Drainage in the pen: 0: no 1: yes; Isolation: Isolation of an individual sheep by panic: 0: no; 1: yes; Housing: How many animals does it take to congregate to avoid panic?; Frecclean: What is the cleaning frequency of sheep housing?; Excreta: What handling of excreta do you carry out in the housing pen?; Brucella ovis_1: 0: negative; 1: positive; Brucella abortus_1: 0: negative; 1: positive; Bratislava_1: 0: negative; 1: positive; Pyrogenes_1: 0: negative; 1: positive; Grippotyphosa_1: 0: negative; 1: positive; Canicola_1: 0: negative; 1: positive; Pomona_1: 0: negative; 1: positive; Tarassovi_1: 0: negative; 1: positive; Hardjo_1: 0: negative; 1: positive; Icterohaemorrhagiae_1: 0: negative; 1: positive; Portland_1: 0: negative; 1: positive; Zone_1: Elevations up to 2600 masl; Zone_2: Elevations between 2601 and 2700 masl; Zone_3: Elevations >2800 masl; Purchased1: Rural market; Puchased2: Imported animals; Purchased3: Born in the flock; Municipality_1: Xalatlaco; Municipality_2: Santiago Tianguistenco; Municipality_3: Calpulhuac; Municipality_4: Chapultepec municipality; Municipality_5: Texcalyacac; Municipality_6: Metepec; Municipality_7: Ocoyoacac; Municipality_8: Lerma; Breed1: Pelibuey; Breed2: Hampshire; Breed3: Suffolk-Pelibuey crossbreed; Breed4: Hampshire-Pelibuey crossbreed; Breed5: Suffolk; Dual_infection1: *Leptospira* spp.-*Chlamydia abortus*; Dual_infection2: smooth *Brucella* spp.-*Brucella ovis*; Dual_infection3: *Leptospira* spp.-smooth *Brucella* spp.; Dual_infection4: *Leptospira-Brucella ovis*; Dual_infection5: *Chlamydia abortus-*smooth *Brucella* spp.; Dual_infection6: *Chlamydia abortus-Brucella ovis*; Mixed_infections1: *Leptospira* spp.-*Chlamydia abortus-*smooth *Brucella * spp.; Mixed_infections2: *Leptospira-Chlamydia-Brucella ovis*; Mixed_infections3: *Leptospira* spp.-*Chlamydia abortus*-smooth *Brucella* spp.-*B. ovis*.

**Figure 4 animals-13-02955-f004:**
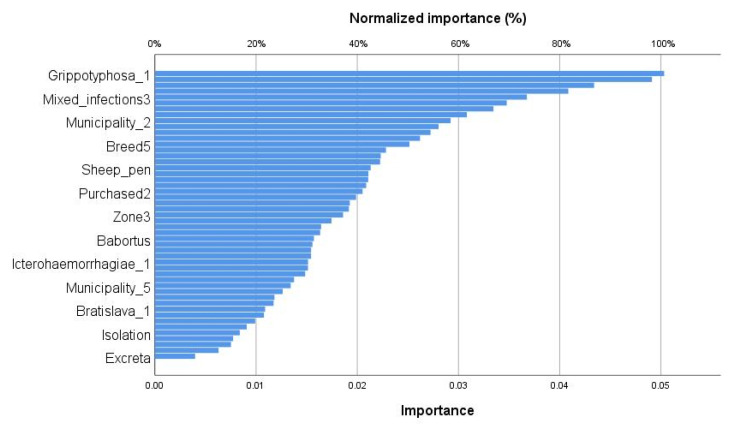
The relative importance of input layer to the prediction for unidentified abortion in ewes. The importance chart shows the relative contribution of predictors that are dominated by Grippotyphosa serovar, followed by Canicola serovar (Portland-vere strain) was detected exclusively, with other infection predictors following far behind. The importance of the mixed infections is clearly visible in infectious abortion among predictors on *x*-axis.

**Table 1 animals-13-02955-t001:** The geographic distribution of weighted seroprevalence of Leptospirosis, Brucellosis (smooth *Brucella* spp., *B. ovis*), and ovine enzootic abortion in the ewe population sampled (by altitudinal zonation).

Microorganisms	Status	Zone 1 (Slopes with Elevations up to 2600 masl)		Zone 2 (Slopes with Elevations between 2600 and 2800 masl)		Zone 3 (Slopes with Elevations above 2800 masl)	
		Prev. (%)	95% CI	Prev. (%)	95% CI	Prev. (%)	95% CI
*Leptospira* spp.	Abortion	38.6	32.9–44.3	13.7	9.7–17.7	5.3	2.7–7.9
	No abortion	36.7	24.5–48.9	18.3	8.5–28.1	5	0.1–10.5
Smooth *Brucella* spp. *	Abortion	48.4	42.6–54.2	18.6	14.1–23.1	7	4.0–10.0
	No abortion	75.0	64.0–86.0	26.7	15.5–37.9	5	0.1–10.5
*B. ovis*	Abortion	6.7	3.8–9.6	5	2.5–7.5	0	N.D.
	No abortion	6.7	3.8–9.6	1.7	0.1–3.2	0	N.D.
*C. abortus*	Abortion	17.5	13.1–21.9	4.6	2.2–7.0	0.7	0.1–1.6
	No abortion	8.3	1.3–15.3	10	2.4–17.6	5.0	0.1–10.5
*Leptospira-C. abortus* dual reactivity	Abortion	15.6	11.4–19.8	10.4	2.6–18.1	6.7	3.8–9.6
	No abortion	16.7	7.3–26.1	14.3	5.4–23.2	20	9.9–30.1
*Leptospira-*smooth *Brucella* spp. dual reactivity	Abortion	55.1	49.3–60.9	50.7	44.3–55.9	50	44.2–55.8
	No abortion	66.7	54.8–78.6	47.6	35–60.2	40	27.6–52.4
*Leptospira-B. ovis* dual reactivity	Abortion	5.3	2.6–7.9	0.7	0.67–0.73	0	N.D.
	No abortion	6.7	0.4–13	1.7	0.1–4.9	0	N.D.
Smooth *Brucella* spp.-*B. ovis* dual reactivity	Abortion	5.6	2.9–8.3	0.7	0.67–0.73	0	N.D.
	No abortion	6.7	0.4 - 13	1.7	0.1–4.9	0	N.D.
Smooth *Brucella* spp.-*C. abortus* dual reactivity	Abortion	12.3	8.5–16.1	3.5	1.4–5.6	0.7	0.67–0.73
	No abortion	8.3	1.3–15.3	5	0.1–10.5	1.7	0.1–4.9
*B. ovis*-*C. abortus* dual reactivity	Abortion	1.1	0.1–2.3	0	N.D.	0	N.D.
	No abortion	0.4	0.1–8.9	0	N.D.	0	N.D.
*Leptospira*-smooth *Brucella* spp.-*B. ovis*-*C. abortus* multi-reactivity	Abortion	0.7	0.67–0.73	0	N.D.	0	N.D.
	No abortion	1.7	0.1–4.9	0	N.D.	0	N.D.

* Kruskal–Wallis test (*p* < 0.03) with uncertainty coefficient 0.02. masl., meters above sea level. Prev., Seroprevalence. CI, Confidence interval. N.D., Not determined.

**Table 2 animals-13-02955-t002:** Weighted seropositivity of *Leptospira* serovars in ewes with history of abortion (by altitudinal zonation).

Serovars (Type Strain)	Status	Zone 1 (Slopes with Elevations up to 2600 masl)		Zone 2 (Slopes with Elevations between 2600 and 2800 masl)		Zone 3 (Slopes with Elevations above 2800 masl)	
		Prev. (%)	95% CI	Prev. (%)	95% CI	Prev. (%)	95% CI
Icterohaemorrhagiae (Palo Alto)	Abortion	37.2	31.5–42.9	12.3	8.5–16.1	5.3	2.7–7.9
	No abortion	20	9.1–30.9	10	1.8–18.2	2	0.1–5.8
Bratislava (Jez-Bratislava)	Abortion	27	21.8–32.2	8.4	5.2–11.6	3.5	1.3–5.6
	No abortion	14	4.5–23.4	8	0.6–15.4	2	0.1–5.8
Canicola (Hond Utrecht IV)	Abortion	12.3	8.5–16.1	4.2	2.5–7.5	3.2	1.1–5.2
	No abortion	10	1.8–18.2	3	0.01–7.6	1	0.1–3.7
Tarassovi (Mitis Johnson)	Abortion	10.2	6.7–13.7	2.5	2.2–7.0	1.4	0.02–2.8
	No abortion	9	1.2–16.8	3	0.01–7.6	2	0.1–5.8
Pyrogenes (Salinem)	Abortion	3.5	1.4–5.6	1.8	2.6–18.1	0.4	0.1–1.1
	No abortion	4	0.01–9.3	0	N.D.	0	N.D.
Pomona (Pomona)	Abortion	2.8	0.8–4.7	1.1	0.1–2.3	0.4	0.1–1.1
	No abortion	4	0.01–9.3	2	0.1–5.8	0	N.D.
Canicola (Portland-vere)	Abortion	2.5	0.7–4.3	0.7	0.67–0.73	0.4	0.1–1.1
	No abortion	2	0.1–5.8	0	N.D.	0	N.D.
Hardjo (H-89)	Abortion	6	0.4–3.8	2	0.67–0.73	0	N.D.
	No abortion	0	N.D.	0	N.D.	5	0.1–10.9
Grippotyphosa * (Moskva V)	Abortion	1.4	0.02–2.8	1.4	0.02–2.8	0.4	0.1–1.1
	No abortion	2	0.1–5.8	0	N.D.	2	0.1–5.8

* Kruskal–Wallis test (*p* < 0.05) with uncertainty coefficient 0.02. masl, meters above sea level. Prev., Seroprevalence. CI, Confidence interval. N.D., Not determined.

**Table 3 animals-13-02955-t003:** Final generalized linear model to estimate the effect of four infectious pathogenic microorganisms and putative factors on unidentified abortion in sheep.

Covariable	β	S.E.	(95%CI)	*p* Value
Intercept	−4.63	0.42	−20.1–10.8	0.56
Grippotyphosa serovar.	−1.98	0.05	−3.7–−0.27	0.02
Ewe gave birth on a bed of straw.	−4.77	0.02	−7.1–2.46	<0.001
Municipality of Lerma.	−17.31	0.19	−24–10.32	<0.001
Municipality of Santiago Tianguistenco.	−8.57	0.11	−12.4–4.7	<0.001
Hardjo serovar.	18.95	0.03	17.9–20	<0.001
*Leptospira*-*Brucella ovis* co-infection.	−18.27	0.08	−21–15.5	<0.001
The water supply for animals.	7.43	0.08	4.5–10.31	<0.001
Sheep pens constructed with metal sheets and untreated wood.	2.21	0.05	0.29–4.14	0.024
Suffolk breed.	−7.6	0.13	−12.42–2.79	0.002
Slopes with elevations above 2800 masl.	−9.2	0.12	−13.7–4.8	<0.001
Slopes with elevations between 2600 and 2800 masl.	7.9	0.1	4.2–11.7	<0.001
Municipality of Chapultepec.	−8.7	0.12	−12.9–4.5	<0.001
Dirt and concrete pen flooring.	9.6	0.12	5.4–13.9	<0.001
Municipality of Texcalyacac.	−1.8	0.05	−3.6–0.001	0.05
Municipality of Calpulhuac	−16.5	0.19	−23.4–9.53	<0.001
*Brucella ovis*.	17.8	0.07	3–10.29	<0.001
Tarassovi serovar.	−1.4	0.03	−2.5–0.42	0.006
Municipality of Xalatlaco.	29.1	0.11	24.9–33.2	<0.001
Agglomeration of excreta near the housing pen.	6.7	0.1	3.0–10.3	<0.001

GLM coefficients (β) with 95% confidence interval (95%CI). S.E., Standard error.

## Data Availability

The datasets used and/or analyzed during the current study are available from the corresponding author upon reasonable request.

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
