# Peer review of "Modelling the Unidentified Abortion Burden from Four Infectious Pathogenic Microorganisms (Leptospira interrogans, Brucella abortus, Brucella ovis, and Chlamydia abortus) in Ewes Based on Artificial Neural Networks Approach: The Epidemiological Basis for a Control Policy"

_animals, 2023, doi:10.3390/ani13182955_

Round 1
Reviewer 1 Report
Arteaga-Troncoso et al. conducted an interesting study on the epidemilogical analysis on four pathogens in ewes related with abortion. I have no major concerns, but there are quite a few problems needs to be addressed. The authors should re-checked the MS carefully to avoid these problems and sth else.
1. Lines 1-4: the authors should indicated which four pathogens in the title
2. Line 43: changed "4" into "four"
3. Line 160: add a blank after "x"
4. Line 211: there are some fromat problems for the formula
5. Lines 340-344: the legend for Figure 1 should not be italic
6. Line 349: the P shoud be italic
7. Line 363: what's the green line means in Figure 2
8. Line 369: the quarlity for Figure 3 is poor, and needed to be improved. I can not see the words clean. The same as Table S1. Supplementary figures and tables should be submitted as separate file
9. Line 636 and after: as to the reference part, why most references have doi number, but some references do not, such as 1 and 2
Minor editing of English language required
Reviewer 2 Report
The authors aimed to model the significance of some abortifacient pathogens in abortion cases of undiagnosed aetiology.
This is an interesting study that can be published after modification as indicated below.
The model is interesting and the findings, based on this model, are valid (I did a few of the calculations and they added up).
Nevertheless, I am very concerned about the limited array of pathogens that were included in the evaluated model. Of the major pathogens, the authors did not include Toxoplasma gondii and Coxiella burnetii. Other pathogens might have been included as well, but anyway they are not so important. However, T gondii and C. burnetii are important causal agents of abortion and have been omitted.
The authors must explain this strategy of exclusion and justify the limited number of pathogens into their model.
This should be carried in a separate passage in the discussion and based on that the authors should also indicate the limitations of their study.
This is a major drawback of the study and the authors must address the matter in detail to continue with publication of the manuscript.
Reviewer 3 Report
The objective of the work was to construct a model to account for the diagnosis of cases of abortion that could not be diagnosed.
Major issues.
The introduction can be extended to include further information about ANN, as this is not a topic that readers of this journal would be familiar.
There is no need to describe details of the techniques used for serological screening (subsection 2.4.), but reference to previous publications will suffice.
Results of serological tests should be also presented in the form of a table.
The detailed results included in the main text should be transferred to supplementary material.
In general, the manuscript should be become more ‘light’, as it contains many details that make reading tiresome.
Minor issues
Chlamydia psittaci, not Chlamydophila
The discussion can divided into three sub-section to facilitate reading.
In discussion, please include a passage with similar findings in other parts of the world.
Also, please include a passage of the potential help that can be offered to clinicians.
Overall. The manuscript is interesting, but substantial changes are needed before acceptance.
Round 2
Reviewer 2 Report
The authors provided adequate response to the comments made regarding the possible omission of other abortifacient pathogens from their model.
The added relevant passages in the Discussion, thus improving the manuscript.
Nevertheless, I would expect them to also add a paragraph regarding the issue being a possible limiting factor of their study; that way, they can cover for all possibilities. The authors should consider that mistakes do happen everywhere and previous findings in Mexico may not fully reflect the current situation....
After this improvement, the manuscript can be accepted.
Reviewer 3 Report
The authors have improved the manuscript by taking into account the comments and suggestions previously made.
I am satisfied with these changes and I recommend acceptance of the manuscript.
